# Barriers and Opportunities for HPV Self-Sampling in Underserved Rural Communities: Insights from a Mixed Methods Study

**DOI:** 10.3390/ijerph22050783

**Published:** 2025-05-15

**Authors:** Joyline Chepkorir, Nancy Perrin, Lucy Kivuti-Bitok, Joseph J. Gallo, Deborah Gross, Jean Anderson, Nancy R. Reynolds, Susan Wyche, Hillary Kibet, Vincent Kipkuri, Anastasha Cherotich, Hae-Ra Han

**Affiliations:** 1School of Nursing, Johns Hopkins University, Baltimore, MD 21205, USAhhan3@jhu.edu (H.-R.H.); 2Department of Nursing Sciences, University of Nairobi, Nairobi 00100, Kenya; 3Department of Mental Health, Bloomberg School of Public Health, Johns Hopkins University, Baltimore, MD 21205, USA; 4Department of Gynecology and Obstetrics, Johns Hopkins University School of Medicine, Baltimore, MD 21287, USA; 5Department of Media and Information, Michigan State University, East Lansing, MI 48824, USA; 6Department of Global Health, Masinde Muliro University of Science and Technology, Kakamega 50100, Kenya; 7Department of Reproductive, Maternal, Newborn and Child Health, GoldStar Kenya, Nairobi 00208, Kenya; 8Department of Public Health, Masinde Muliro University of Science and Technology, Kakamega 50100, Kenya; 9Department of Health, Behavior, and Society, Johns Hopkins University Bloomberg School of Public Health, Baltimore, MD 21205, USA

**Keywords:** cervical cancer, screening, health information, sub-Saharan Africa, barriers, facilitators, HPV-DNA testing, self-sampling, willingness, rural

## Abstract

Cervical cancer is the leading cause of cancer-related deaths among women in sub-Saharan Africa, especially in rural areas with limited access to screening. This study explored factors influencing rural Kenyan women’s willingness to self-collect samples for HPV-DNA testing. Data were drawn from a mixed methods study in two Kenyan rural counties, including surveys with 174 women and interviews with 21 participants. The mean age of the survey sample was 45.2 (SD = 13.2) years. Only 6.4% had ever been screened, yet 76.9% expressed willingness to self-collect samples for testing. Increased willingness was associated with cervical cancer awareness (OR = 3.49, 95% CI = 1.50–8.11), relying on health workers as primary sources of health information (OR = 1.88, CI = 1.23–2.86), or the news media (OR = 2.63, CI = 1.27–5.48). High cervical cancer stigma (OR = 0.71, CI = 0.57–0.88) and longer travel times of 30–120 min to a health facility (OR = 0.44, CI = 0.20–0.93) were linked to reduced willingness. Integration of the findings showed that comprehensive health promotion—through education, health worker endorsement, and mass media campaigns—may improve HPV self-sampling uptake and reduce the cervical cancer burden in rural Kenya.

## 1. Introduction

Cervical cancer (CC) is the most common cancer among women in sub-Saharan Africa (SSA), particularly among women aged 21 to 48 years [1,2]. In Kenya, CC is the leading cause of cancer-related deaths, with about 5236 new cases diagnosed annually and over 3200 deaths in 2020 [3]. Over 90% of women are diagnosed at advanced stages despite prior contact with the healthcare system [4]. About 63% of invasive CC cases in the country are caused by high-risk human papillomavirus (HPV) types 16 and 18, which can be prevented through HPV vaccination [5]. In 2019, the quadrivalent, two-dose HPV vaccine Gardasil-4 was introduced, targeting adolescent girls and young women in schools and health facilities [6,7]. Unfortunately, the HPV vaccine coverage remains low, as only 31% of the targeted group had received the two doses of HPV vaccine by 2021 [6].

CC screening is a promising strategy to detect CC among an estimated 13 million Kenyan women aged 15 years and older who are at risk for the disease [8]. Kenya’s National Cervical Cancer Screening Guidelines have been implemented for over a decade and currently recommend the screening of all eligible women aged 30 to 49 years [3]. As part of the 2030 global cancer elimination strategy, the World Health Organization (WHO) recommends a target screening of 70% of eligible women and treatment of 90% of those diagnosed with CC in each country [9]. However, the screening uptake rate is approximately 16 to 18% in screening-eligible Kenyan women aged 18 to 69 years, and the rates are notably lower in rural areas [4]. The WHO and Kenyan cancer screening guidelines recommend HPV-DNA testing as the primary screening method since it has a high sensitivity to detect high-risk HPV sub-types [10,11].

In Kenya, over 90% of health facilities use visual inspection methods and a few of them use Pap smears and HPV testing, which are often constrained by sociocultural beliefs, limited availability of skilled workforce, and lack of essential infrastructure [12]. The self-sampling approach for HPV-DNA testing offers an opportunity to circumvent these barriers by paving the way for community-based screening and the increased reach and empowerment of women to collect specimens in private spaces at convenient places and times without the fear of pain associated with pelvic examinations used in other procedures [12]. Nonetheless, the success of self-sampling depends on the target women’s acceptance [13]. Although studies in SSA have affirmed a high acceptability of self-sampling in diverse settings, few studies have investigated the acceptability of this procedure in rural community settings.

Empirical studies on the use of self-sampling in rural communities in Kenya are rare. In a qualitative study conducted among rural Kenyan women (N = 120), women reported generally positive experiences with HPV self-sampling. In an urban sample (N = 409), more than 80% of women reported that they would be comfortable using a self-sampling device and 84% would prefer at-home sample collection [13]. Similarly, another study among rural Kenyan women (N = 97) found that 90% of them were willing to collect their samples in private places [8]. As HPV testing and self-collection kits gradually become available in Kenya’s healthcare system, it is critical to understand the factors that may impact women’s willingness to self-collect samples for HPV-DNA testing [10].

This study’s aim was to investigate potential barriers to and facilitators of self-sampling willingness in order to inform the design and implementation of interventions aimed at promoting women’s adoption of self-sampling for HPV-DNA testing. To the best of our knowledge, factors associated with HPV-DNA self-sampling willingness have not been investigated in rural, community-based contexts in Kenya and other parts of SSA. To this end, the research question was as follows: what individual, interpersonal, organizational, community, and structural factors influence rural Kenyan women’s willingness to self-collect samples for HPV-DNA testing for CC screening? We hypothesized that

(1)Women who independently make healthcare decisions are more likely to express willingness to self-collect for HPV-DNA testing than those whose significant others make decisions for them;(2)Prior exposure to CC information via news media (TV/radio) and interpersonal sources (e.g., health workers, social networks) is positively associated with willingness to self-collect for HPV-DNA testing;(3)Higher levels of anticipated CC stigma are negatively associated with willingness to self-collect;(4)Women who live closer to a health facility (shorter travel time) are more likely to express willingness to self-collect for HPV-DNA testing than those who travel longer distances.

## 2. Materials and Methods

### 2.1. Theoretical Framework

This study was guided by the socio-ecological framework, which proposes that an individual’s behavior affects and is affected by the social environment [14]. It emphasizes the need for health interventions to target changes at the individual/intrapersonal, interpersonal, organizational/institutional, community, and policy levels to support and maintain healthy behavior [14]. We presupposed that at the individual level, an individual’s willingness to screen might be influenced by their capacity to make healthcare decisions, access to resources (i.e., health insurance), and their awareness of CC (i.e., need for screening) [15]. At the interpersonal level, we theorized that an individual’s self-sampling willingness could be influenced by interactions with their peers and household dynamics (i.e., spouse’s involvement in a woman’s healthcare decision making) [16]. At the organization level, we presumed that health workers and health facilities could potentially impact a woman’s willingness to self-collect samples for HPV testing [17]. At the community level, we hypothesized that factors such as beliefs and CC stigma might influence self-sampling willingness [18]. Lastly, we examined whether health system and policy-level factors such as the geographical location of health facilities have the potential to influence successful self-sampling [16,17].

### 2.2. Study Design and Setting

This study was a secondary analysis of data from a parent convergent mixed methods study, which explored the interplay of health information, health literacy, and CC screening uptake. The study used interviewer-administered surveys and semi-structured interviews to collect data between August and September 2023 in rural Bomet and Kericho counties, Kenya. Kenya is subdivided into 47 counties. Bomet and Kericho counties, located in Kenya’s South Rift of the Great Rift Valley, consist of predominantly rural populations, 96.8% and 89.6%, respectively [19]. These counties are adjacent and consist primarily of Kipsigis communities that have comparable cultural and economic practices.

### 2.3. Study Sample

Women were recruited from community settings. Convenience sampling was used to select voluntary participants for the quantitative arm who were available and met the inclusion criteria. Eligibility criteria were as follows: (1) women aged 18 to 65 years (age range recommended for CC screening based on Kenya’s guidelines), (2) proficient in Kipsigis language, (3) attained some primary school education (between grade 1 and 8), and (4) a resident of Bomet County or Kericho County. Participants were excluded if they were either (1) acutely/terminally ill or (2) were cognitively impaired, limiting participation in study activities. A total of 174 eligible women participated in the survey study. Purposive maximum variation sampling was used to select a sub-sample of participants *(n* = 21) with varied characteristics for semi-structured interviews [20]. Specifically, CC screening status (ever versus never screened), CC awareness (ever versus never heard), between ages of 35 and 45 years (common ages for CC diagnosis), and anticipated CC stigma (none versus at least one) were used as the basis for selection for qualitative interviews.

### 2.4. Procedures

The survey, interview guide, and consent form were originally developed in English, translated to Kipsigis, and then back translated by experts to ascertain the accuracy of the translation [21]. Two bilingual registered nurses from each county and one medical doctor specializing in obstetrics and gynecology reviewed the survey and interview guide for face validity and contextual relevance. Based on their feedback, some questions that were not contextually relevant were removed. The survey was pre-tested among the first 24 women while the interview guide was pilot tested among 10 women. Amendments to the survey and interview guide were minor and mainly involved the re-phrasing of questions for contextual aptness.

Five trained research staff residing in the two counties recruited study participants, using study flyers and word of mouth, from participants’ homes, local health centers, shopping centers, and churches. The interviewer-administered survey and interviews were offered in convenient private spaces of the participants’ choosing (e.g., inside or outside their house). The survey was administered using REDCap software version 13.2.0 in offline mode, due to limited access to the Internet, and immediately transferred to the web after data collection. All interviews were audio-recorded to facilitate verbatim transcription and analysis. To maintain participants’ confidentiality, all audio recordings were exported to a secure platform before being deleted from the original devices. Each survey and interview took approximately 30 to 45 min. Participants were incentivized with Ksh.330 (approximately USD 2.37) for responding to the survey and additional Ksh.330 for those who were selected for interviews. Upon completion of data collection, transcripts and survey data were checked for accuracy and completeness and cleaned.

#### Ethics

Study procedures were approved by the Johns Hopkins Medicine Institutional Review Board (IRB00357410). Additionally, in line with the local requirements for conducting human subject research, study procedures were reviewed by Amref Ethical and Scientific Review Committee in Kenya (P1394/2023) and licensed (Ref No. 774947) by the Kenya National Commission for Science, Technology and Innovation. The research staff obtained verbal consent in Kipsigis language from each participant before the study activities. The consent document detailed information about the study’s purpose and risks and benefits of participating. Each participant could ask questions about the study procedures before consenting.

### 2.5. Measures

#### 2.5.1. Independent Variables

Factors at the individual level assessed were demographics including age (in years), marital status (single, married, separated, divorced), education (no formal education, lower primary education [grades 1–3], upper primary [grades 4–8]), employment (self-employed, unemployed, employed in private sector), perceived health status (poor, fair, good, very good), and CC awareness (ever/never heard about CC). Additionally, resources considered were household monthly income $(≤ to 35, 36–70, 71–106, 107–142, 143–178, 179–214, 215–251, 252–322), income comfort level (comfortable or not comfortable), and health insurance status (insured or uninsured); primary sources of health information (news media, social network, community leaders, health workers, teachers and herbalists); and estimated travel time to the nearest health facility (in minutes). Each of these variables were included in the study questionnaire as individual items.

Factors considered at the interpersonal level in this study were as follows: healthcare decision making (self, self and spouse, spouse only, or mother), interpersonal sources of CC information (social networks, community leaders, health workers, teachers), and interpersonal sources of general health information (health workers and social networks). Participants could select more than one response. At the organizational level, factors examined were the reliance on healthcare workers as primary sources of health information (doctors, nurses, and community health workers [CHWs]), receiving CC-related information from healthcare workers (whether they had ever heard about CC from health workers), and prior CC screening (ever screened or never screened). Factors assessed at the community level were the reliance on news media (TV and radio) as primary sources of health information, having heard about CC information from the news media, and anticipated CC stigma. The 8-item CC stigma scale was used to determine anticipated CC stigma [18]. The internal consistency reliability of this instrument was 0.90 [18] in the original study and 0.83 in this study. Lastly, at the health system and health policy levels, the ranking of the nearest health facility (level 1 to 4) and travel time to the nearest health facility (in minutes) were evaluated.

#### 2.5.2. Outcome Variable: Self-Sampling Willingness

Willingness to self-collect samples for HPV-DNA testing was assessed using a two-part question. The first part of the question explained the self-sampling procedure: “HPV self-sampling allows women to collect their own samples for CC screening using a swab or a brush”. The second part assessed willingness to accept self-collection. “Would you be willing to self-collect a sample for testing if this screening is suitable for you?” Responses to this question were binary (yes/no).

### 2.6. Data Analysis

The analytical sample consisted of participants (*n* = 24) who were involved in the pre-testing of the survey and 150 women who completed the study survey. Analysis of quantitative data used descriptive statistics (means, standard deviations, medians, range, frequencies, and percentages) to summarize the sample characteristics. Also, we used binary logistic regression to assess the correlates of self-sampling willingness. We excluded one survey participant with a missing response on self-sampling willingness; hence, the logistic regression output is based on 173 respondents. Quantitative data analysis was conducted using STATA/BE version 17 software.

Qualitative data were analyzed using Dedoose software version 9.0.5 (https://www.dedoose.com/), in Kipsigis language, by four bilingual coders. We used theoretical thematic analysis to identify codes and generate themes relating to the research study aim [22]. Additionally, we employed inductive analysis by allowing the research aim to evolve through the coding process [22]. Specifically, each transcript was coded by two coders independently and all coders met regularly to resolve any discrepancies and discuss emerging codes. Coded files were exported in Word format. Themes, sub-themes, and accompanying quotes were organized in an Excel spreadsheet before being translated to English by two of the coders. Thereafter, results from each study arm were merged and juxtaposed using a joint display [23].

## 3. Results

Table 1 summarizes the characteristics of the survey sample. The mean age of the survey sample was 45.2 (SD *=* 13.2) years. The majority were from Bomet County (64.4%). More than half (81.5%) of the sample had attained formal education between grades four and eight and the majority were married (83.8%). About 78% of the women were self-employed. Most participants (88.4%) earned USD 35 or less per month in their households. The sample was predominantly uninsured (76%), and the majority (63%) rated their health status as either good or very good.

The nearest health facilities accessed and used by most participants (75.7%) were level 2 (dispensary/clinic) and most of them walked (78.5%) to get services in these facilities. Travel time to the nearest health facilities was between half an hour to two hours for many (45%) of them. Approximately 80% of the participants made their own healthcare decisions.

Regarding health information access, about 69.4% and 70% of participants reported that they primarily accessed health information from health workers and the news media, respectively. Most participants (83.2%) had heard about CC from the news media (37%), their social networks (20.8%), health workers (24.3%), or other sources (1.7%). Only 6.4% of the sample had ever been screened for the disease.

The CC stigma scale showed that half of the participants anticipated between one and eight potential drivers of CC stigma. Only 1.7% of participants perceived they were at risk for CC. Most participants, 76.9%, reported that they would be willing to self-collect samples for HPV-DNA testing if offered. Among women (*n* = 40) who were unwilling to self-collect samples, the majority (72.5%) indicated that they lacked the confidence to collect samples, while others reported that they were not interested (4%), and afraid of either the procedure (3.5%) or the results (0.6%). The sample characteristics of the sub-sample of participants interviewed are summarized in Appendix A: Table A1.

### 3.1. Factors Associated with Self-Sampling Willingness

The bivariate logistic regression outputs of sample characteristics on self-sampling willingness are summarized in Table 2.

The results indicate that those who had heard about CC were 3.49 times more willing to self-collect samples than those who had not (odds ratio [OR] = 3.49, 95% confidence interval [CI] = 1.50–8.11). Respondents whose sources of CC information were the news media (radio and television) were 143% more likely to accept self-sampling if offered (OR = 2.43, 95% CI = 1.07–5.51). Using the news media and health workers as primary sources of health information was associated with 163% (OR = 2.63, 95% CI = 1.27–5.48) and 88% (OR = 1.88, 95% CI = 1.23–2.86) higher odds of self-sampling willingness. On the other hand, CC stigma was associated with 30% lower odds (OR = 0.71, 95% CI = 0.57–0.88) of self-sampling willingness. Traveling for half an hour to two hours, compared to less than half an hour, to the nearest health facility was associated with 56% lower odds (OR = 0.44, CI = 0.20–0.93) of self-sampling willingness. Healthcare decision making was not significantly associated with women’s willingness to self-collect samples for HPV-DNA testing.

### 3.2. Qualitative Findings

The qualitative findings were mainly congruent with the quantitative results. Nonetheless, we identified some discrepancies in the qualitative results; some participants who had heard about CC from the news media were unwilling to self-collect samples for HPV testing. Additionally, while most participants anticipated CC stigma from their community, some participants reported that stigmatizing attitudes from people may be buffered by Christian religious beliefs. Below are the key themes and example quotes.

**Cervical cancer awareness:** Women who were open to self-sampling often indicated that their willingness to self-collect was influenced by their understanding of the importance of cervical cancer screening. Some participants who expressed interest in self-sampling also emphasized the need for additional education on the procedure before collecting samples. One respondent in her 60s shared the following:
“*I would agree to self-collect a sample for screening because I heard that screening helps detect cervical cancer early*.”

**Acquisition of health information from health workers:** Respondents who primarily received health information from healthcare workers such as doctors, nurses, and CHWs emphasized their critical role in promoting knowledge about cervical cancer screening. Trust in healthcare providers, along with their recommendations, emerged as key facilitators of self-sampling. One participant in her 40s explained her willingness to self-collect for HPV-DNA testing:
“*I would not refuse especially if a healthcare provider advised it, because I also want to know my health status*.”

**The role of news media sources**: The qualitative data revealed that news media play a key role in promoting cervical cancer knowledge and may influence willingness to self-collect samples. One participant in her 30s, who had heard about cervical cancer, noted that she became aware of the importance of screening through an educational radio program—her primary source of health information:
“*I heard a talk about cervical cancer on the radio. Otherwise, I haven’t seen anyone with the disease; I’ve only heard about it.*”

However, there were some women who were unwilling to self-collect samples for screening despite demonstrating awareness of screening through news media. A participant in her 40s acknowledged that she understood the importance of screening but expressed reluctance to self-collect due to fear:
“*I heard about cervical cancer from an educational program on the radio. They announced that women should go to the hospital for screening. The radio is my main source of information—they often teach and emphasize the importance of getting screened. Personally, I don’t think I would want to screen… I’m afraid of the procedure.*”

**Anticipated cervical cancer stigma:** Several women from both counties anticipated cervical cancer stigma in their communities, primarily due to the symptoms of the disease. However, some participants suggested that religious beliefs could help reduce instances of stigma. For example, a participant in her 50s shared that cervical cancer symptoms might lead a woman’s spouse to infidelity, causing the woman distress and self-blame:
“*Cervical cancer might cause a woman to have a foul-smelling discharge. Her husband might reject her and seek another woman, which could cause her emotional distress, as she may feel the disease led to infidelity. Some men may al so reject their wives after a diagnosis due to reduced sexual intimacy.*”

In contrast to reports of stigma, some women noted that individuals with religious beliefs might be less likely to stigmatize a woman diagnosed with cervical cancer. A participant in her 50s explained that responses to a diagnosis could vary based on belief systems:
“*Friends who are religious may offer her hope, while those with more secular beliefs might suggest she developed the disease due to infidelity.*”

Longer travel time to the nearest health facility: Several participants reported that they had not undergone CC screening due to the inability to afford transportation to health facilities and the lack of screening equipment at nearby centers. Some noted that even accessible facilities lacked the necessary resources for screening. One participant shared the following:
“*Nothing prevented me from going for screening except transportation costs. There was a free mass screening campaign, but it was announced just a day before, and I didn’t have the money to cover transport.*”

Some participants disclosed that their nearest health facilities were inadequately equipped to provide CC screening:
“*The health facility is poorly equipped—there’s no screening equipment or machines available for cervical cancer screening.*”

### 3.3. Integration

Table 3 presents a joint display integrating the mixed methods findings. For each measure, we offer evidence-based recommendations to support the successful adoption of self-sampling for HPV-DNA testing. These recommendations highlight the need to (1) emphasize the significance of CC screening during CC awareness campaigns, (2) optimize patient–provider interactions to promote HPV testing via self-sampling, (3) promote health-worker-led CC information dissemination, (4) leverage mass media to boost CC education and screening efforts, (5) complement radio broadcasts of CC education and screening with health-worker-led interventions, (6) encourage discussions about CC in communities to debunk stigma, (7) explore the potential mediating role of religious leaders in reducing CC stigma, (8) increase awareness of CC symptoms and available treatment options, (9) establish support groups for women diagnosed with CC, and (10) provide HPV testing kits in community settings to promote access and increase screening uptake among eligible women.

## 4. Discussion

We sought to investigate potential barriers and facilitators of rural Kenyan women’s willingness to adopt self-sampling for HPV-DNA testing. The sample predominantly comprised women with low CC screening uptake, most of whom expressed willingness to self-collect samples for CC screening if offered. Among the factors examined, CC awareness, receipt of general health information primarily from healthcare providers and news media, and having heard of CC from the news media were positively associated with willingness to self-sample. In contrast, anticipated CC stigma and travel times of 30 to 120 min to the nearest health facility were negatively associated with willingness. Overall, 76.9% of the participants indicated they would self-collect samples for CC screening if given the opportunity. This is consistent with findings from a longitudinal study that sampled rural Malawian women (N = 122), where 66% expressed willingness to self-collect samples for HPV testing [24]. However, that study also revealed a gap between hypothetical willingness and actual uptake: only 53% and 47% of those who were willing and unwilling at baseline, respectively, went on to self-collect samples 12 to 18 months later [24]. While both studies suggest a generally high level of willingness to self-sample, they underscore the importance of identifying and addressing the factors that influence actual uptake of HPV self-sampling.

CC awareness and women’s willingness to screen for CC are critical for the prevention of HPV and CC [25]. In this study sample, 83.2% of the participants had heard of CC, but only 6.4% had ever been screened in their lifetime. These findings closely mirror results from a Tanzanian sample, where 82% of women were aware of CC, yet only 6% had ever been screened for precancerous lesions. CC awareness was significantly associated with 249% increased likelihood of self-sampling willingness in our sample. These results highlight the need to expand access to information about CC and self-sampling. Previous research has shown that awareness of screening methods and knowledge of where to screen are critical for women’s uptake of CC screening services [21,26]. The qualitative findings from this study further demonstrate that primary prevention through education—specifically, providing information about the importance of CC screening and demonstrating the self-sampling procedure—are key facilitators of women’s willingness to self-sample. Comparably, researchers investigating a sample of Ugandan women found higher self-sampling acceptability when providers educated participants and allowed them to examine the collection brush prior to self-collection [27]. Consistent with prior research, rural women in sub-Saharan Africa tend to have substantially lower levels of CC awareness and knowledge compared to their urban counterparts, largely due to limited access to health information [21,28]. To promote uptake of HPV-DNA self-sampling, CC education—particularly that which includes practical, hands-on instructions for self-collection—will be essential.

Our findings highlight the instrumental role of the traditional news media—particularly radio—in enhancing HPV-DNA testing. Women who identified news media as their primary source of general health information, and those who had heard about cervical cancer through these channels, were more likely to express willingness to self-collect samples. These results align with previous research showing that women in sub-Saharan Africa often learn about cervical cancer from television and radio [29,30,31,32]. For instance, among rural Tanzanian women, regular radio listeners had approximately 25% higher odds of undergoing screening [33]. While news media are critical for disseminating cervical cancer information, our qualitative findings indicate that low self-efficacy related to self-collection may hinder some women who are otherwise willing to be screened. It is therefore essential for stakeholders to assess and strengthen women’s understanding of the self-sampling procedure, particularly among those who express willingness to participate.

Healthcare workers (HCWs) play a pivotal role in disseminating information about CC and CC screening. In this study, women who primarily received health information from HCWs were 88% more likely to express willingness to self-collect samples for HPV-DNA testing. The qualitative findings further revealed that many women first learned about CC from HCWs, and several stated they would be more likely to accept self-sampling if recommended by a healthcare provider. Although we did not find a statistically significant association between having heard about CC from HCWs and self-sampling willingness, prior studies have shown that receiving information from healthcare providers is significantly associated with higher odds of screening uptake [31,34]. Given that nurses often staff rural health clinics frequented by most participants in this study, they are well-positioned to provide CC education both within health facilities and through community outreach in more remote areas. To be effective, CC education should be culturally sensitive, clearly explain the screening process—including self-sampling procedures—and emphasize the importance of early detection in improving health outcomes [28].

Prior research has established that cervical cancer (CC) stigma is a significant barrier to symptom disclosure, seeking screening services, and accessing treatment in many sub-Saharan African communities [13,35,36,37]. In this study, the anticipated CC stigma was notably high, with 57.8% of participants anticipating at least one form of CC stigma. This proportion is substantially higher than that reported in another rural Kenyan sample (n = 419), where only 20.3% of women anticipated CC stigma [35]. The difference may be attributed to sample characteristics: the previous study recruited women from health facilities, 55.6% of whom were HIV positive, while our study primarily recruited women from community settings, with only 2.5% being HIV positive. Women living with HIV may have more frequent interactions with healthcare providers, which can provide opportunities for support and education, thereby reducing stigma. To promote CC screening in rural communities, it is critical to address and reduce stigma both in community settings and within health facilities. Interventions should focus on normalizing conversations around CC, fostering supportive environments, and leveraging trusted community and religious leaders to challenge misconceptions and encourage screening uptake.

A longer travel time to the nearest health facility was significantly associated with a 56% decrease in the odds of willingness to accept self-sampling in this study. This finding aligns with previous research from various sub-Saharan African (SSA) countries, which has shown that long travel distances and associated transportation costs are key barriers to cervical cancer (CC) knowledge and screening [37,38,39]. For instance, in a pooled analysis of 40,555 women from demographic and health surveys conducted between 2013 and 2021 in Benin, Côte d’Ivoire, Cameroon, Kenya, and Namibia, 62.4% of women who reported distance to health facilities as a major barrier were rural residents [39]. Among those who cited distance as a significant issue, only 8% (n = 12,899) had ever been tested for CC, compared to 13.5% of women who did not perceive distance as a barrier. The qualitative findings in our study supported the quantitative results, revealing that the high cost of transportation to better-equipped health facilities was a major concern among participants. This echoes findings from other studies conducted in Kenya and across SSA [13,21,37]. In our sample, level 2 health facilities—financed by county governments—were the most accessible. Although these facilities typically offer free CC screening using visual inspection techniques conducted by nurses, frequent stockouts of screening supplies often force women to seek services at more advanced, distant facilities [35]. To improve uptake of HPV-DNA self-sampling, ministries of health in Kenya and similar SSA contexts should ensure that self-sampling kits are consistently available and easily accessible in rural health facilities.

### Limitations

This study has several limitations. First, participants were selected through convenience sampling, which may introduce selection bias. Second, data collection was limited to rural women from two counties, limiting the generalizability of findings to other settings. Third, all of the data were self-reported, raising the potential for recall bias that could lead to the under- or over-reporting of certain variables. Lastly, some discrepancies emerged between the qualitative and quantitative findings, indicating the need for further research to validate our results.

Despite these limitations, a key strength of this study lies in its use of both qualitative and quantitative data. This mixed methods approach enabled the integration of findings and provided a more comprehensive understanding of the multi-level factors influencing women’s willingness to self-collect samples for HPV-DNA testing.

## 5. Conclusions

Self-sampling for HPV-DNA testing represents a promising approach to cervical cancer (CC) screening and early detection among rural Kenyan women. To support its successful adoption, targeted CC education—delivered through trusted channels such as news media and healthcare workers—will be essential for increasing knowledge, reducing stigma, and improving women’s willingness to self-collect samples. Additionally, addressing structural barriers and improving access to screening services for medically underserved rural populations are critical. Contrary to our hypothesis, healthcare decision-making autonomy was not significantly associated with willingness to self-collect, suggesting that other factors—particularly access to information and structural challenges—may have a greater influence on screening behaviors, regardless of who holds decision-making power within the household.

## Figures and Tables

**Table 1 ijerph-22-00783-t001:** Sample characteristics (*n* = 174).

Characteristic	*n* (%)/Mean (SD)/ Median (Range)
Age (Mean [SD])	45.3 (13.2)
Marital status	
Married	145 (83.3%)
Unmarried	29 (16.7%)
County of residence	
Bomet	112 (64.4%)
Kericho	62 (35.6%)
Education	
No formal education	32 (18.4%)
Lower primary (grades 1–3)	52 (29.9%)
Upper primary (grades 4–8)	90 (51.7%)
Employment status	
Self-employed	134 (77%)
Unemployed	34 (19.5%)
Employed in private/public sector	5 (2.9%)
Missing	1 (0.6%)
Household monthly income	
≤USD 35	154 (88.5%)
USD 36–142	20 (11.5%)
Insurance status	
Insured	41 (23.6%)
Uninsured	131 (75.3%)
Missing	2 (1.1%)
Health status	
Poor/Fair	65 (37.4%)
Good/Very good	109 (62.6%)
Nearest Health Facility	
Level 2 (dispensary/clinic)	132 (75.9%)
Level 3 (health center)	23 (13.2%)
Level 4 and 5 (county hospital/referral)	19 (10.9%)
Transportation to nearest health facility	
Walk	136 (78.2%)
Motorcycle	35 (20.1%)
Public vehicle	2 (1.2%)
Missing	1 (0.6%)
Travel time to nearest health facility	
<30 min (reference)	76 (45%)
30–120 min	93 (55%)
Missing	4 (2.3%)
Healthcare decision making	
Self	139 (79.9%)
Self and Spouse	12 (6.9%)
Mother or spouse	7 (4%)
Missing	16 (9.2)
Primary sources of health information *	
News media (TV and radio)	120 (69%)
Social networks and community	64 (36.8%)
Health workers	121 (69.5%)
Other (herbalist, teachers)	2 (1.1%)
Sources of CC information *	
News media (TV and radio)	64 (37%)
Social networks	36 (20.8%)
Healthcare workers	42 (24.3%)
Other (teachers, religious leaders)	3 (1.7%)
CC awareness	
Ever heard of CC	144 (82.8%)
Never heard of CC	29 (16.7%)
Missing	1 (0.6%)
CC screening status	
Never screened	163 (93.7)
Ever screened	11 (6.3%)
Anticipated CC stigma	
Yes	101 (58.1%)
No	73 (42%)
Willingness to self-sample for HPV DNA testing	
Yes	133 (76.4%)
No	40 (23%)
Missing	1 (0.6%)

Key: SD–standard deviation, * Responses to multiple-choice items (select all that apply), HPV DNA—Human Papillomavirus Deoxyribonucleic Acid, CC–cervical cancer.

**Table 2 ijerph-22-00783-t002:** Bivariate analysis of factors associated with self-sampling willingness (*n* = 173).

Characteristic	OR	*p*-Value	CI
Age (Mean (SD))	0.98	0.14	0.95–1.01
Marital status	0.68	0.47	0.24–1.93
Education	1.06	0.78	0.68–1.68
Employment Status	1.04	0.91	0.49–2.24
Income	0.94	0.67	0.71–1.24
Comfortability with income	0.57	0.33	0.18–1.78
Insurance status	1.35	0.50	0.56–3.21
Health status	1.71	0.06	0.98–3.00
Healthcare decision making	1.07	0.84	0.51–2.27
Cervical cancer awareness	3.49	0.004 **	1.50–8.11
Prior cervical cancer screening	1.38	0.69	0.29–6.66
Cervical cancer stigma	0.71	0.001 **	0.57–0.88
Nearest health facility	0.96	0.86	0.58–0.59
Distance to nearest health facility			
<30 min (ref)			
30–120 min	0.44	0.032 *	0.20–0.93
Transportation to nearest health facility	0.64	0.24	0.31–1.34
Sources of cervical cancer Information			
News media (TV and radio)	2.43	0.03 *	1.07–5.51
Social networks	0.61	0.24	0.27–1.38
Healthcare workers	1.48	0.41	0.59–3.7
Primary sources of health information			
News media (TV and radio)	2.63	0.01 **	1.27–5.48
Social networks and community	1.06	0.77	0.72–1.55
Health workers	1.88	0.003 **	1.23–2.86

OR—odds ratio, CI—confidence interval, SD—standard deviation, ref—reference. * *p*—value < 0.05, ** *p* value ≤ 0.01.

**Table 3 ijerph-22-00783-t003:** A joint display of the qualitative findings, quantitative findings, and mixed methods meta-inferences.

	Quantitative Findings	Qualitative Findings	Mixed Methods Meta-Inferences
Measure	OR (95% C.I.)	Themes and Quotations	Recommendations for enhancing self-sampling for HPV-DNA testing
Cervical cancer awareness	3.49 (1.50–8.11)	**Knowledge about the significance of screening**: “*I had heard healthcare workers saying that it is necessary for women to get screened so they know if they have the disease and that it is important to get regular screening.*” (between 20 and 25 years old, willing to self-collect a sample).	Emphasize the significance of screening in CC awareness campaigns to promote screening through sample self-collection for HPV-DNA testing.
Acquisition of health information primarily from health workers	1.88 (1.23–2.86)	**Trust in healthcare providers:** “*I trust a doctor or a nurse. You know, they trained in the healthcare field, and they explain things about health and wellbeing very well based on what you tell them. When you discuss your health issue with them, they would give you a detailed explanation about it.*” (between 46 and 50 years old, willing to self-collect a sample, primarily accesses health information from a doctor or a nurse)	Optimize provider–patient interactions to educate patients on the importance of CC screening and offer sample self-collection kits to women who are eligible and willing to screen.
		**Cervical cancer information acquisition from health workers**: “*I have come to know that it (screening) is important because cervical cancer has caused the death of so many women. When health workers came (to our community) to educate people (about cervical cancer screening), I understood that it is important for one to go for screening, so that if you have it (cervical cancer), you can be treated when curative treatment is still possible.*” (heard of CC, between 51 and 55 years old, willing to self-collect samples, primarily accesses health information from a doctor or a nurse).	Promote health-worker-led CC information dissemination to ensure provision of accurate CC information and sample self-collection for HPV-DNA testing among eligible women.
News media sources of cervical cancer information	2.63 (1.27–5.48)	**Television and radio sources**: “*I heard (of cervical cancer information) over the radio.... and watched on television.*” (between 61 and 65 years old, willing to self-collect a sample)	Maximize the potential for these mass media to facilitate CC information dissemination and screening uptake.
		**Radio-based educational programs and announcements**: “*I have heard from the radio...It was said that it affects women, particularly the cervix...It was announced that screening was being done for free at the hospital.*” (between 56 and 60 years old, unwilling to self-collect a sample, primarily obtains health information from the radio, heard of CC)	Complement radio-based CC information dissemination in local dialects with education and support from health workers.
Anticipated cervical cancer stigma	0.71 (0.55–0.88)	***Community stigma****:* “*you know if someone has cervical cancer and they have reached the late stages, there is nothing to hide. It can be concealed when they are still alive but when they die, then you will hear that they had this type of cancer.*” (between 36 and 40 years old, willing to self-collect a sample)	Encourage discussions about CC as a disease to normalize the topic and enhance screening uptake.
		**Stigma and potential mediating role of religion**: “*If it is a Christian family, her spouse might not stigmatize her—he will take care of her. But in a household without salvation, he might stigmatize her and even send her away because of the lack of sexual intimacy.*” (Participant, aged 30–35, willing to self-collect a sample)	Explore the potential role of religious leaders to lead open dialogue about CC and reduce stigma associated with CC.
		**Community stigma:** (a woman with cervical cancer may be stigmatized in the community) “*...because cervical cancer is a bad disease, and it was said that it causes one to have a foul odor.*” (between 45 and 50 years old, willing to self-collect a sample)	Increase awareness of CC symptoms and treatment, and create support groups for women diagnosed with CC.
Longer travel time to the nearest health facility	0.44(0.20–0.93)	**Travel distance barriers:** (I have not gone for screening because the health facility) “*...where screening is being conducted is far.*” (between 51 and 55 years old, willing to self-collect a sample)	Provide self-sampling kits in community-based settings to ensure easy access and screening uptake

## Data Availability

The data supporting the reported results can be found here: http://datadryad.org/share/fjdv_UKmPJbViLbFkEUxegSJFjIcewyuZBFVmPhiK8s (accessed on 10 September 2024).

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
