# Peer review of "Barriers and Opportunities for HPV Self-Sampling in Underserved Rural Communities: Insights from a Mixed Methods Study"

_ijerph, 2025, doi:10.3390/ijerph22050783_

Round 1

Reviewer 1 Report

Comments and Suggestions for Authors

The paper is well-written and needs very minor modifications.

Introduction

Text: CC is the leading cause of cancer-related deaths, with about 5,236 new cases diagnosed annually. 
Comment: suggesting to use number of deaths instead number of new cases since statement pertains to CC as the leading cause of cancer-related mortality

Results

Table 1 - can be placed in supplementary data

Author Response

Summary

Thank you very much for taking the time to review this manuscript. Please find the detailed responses below and the corresponding revisions in the re-submitted file.

Introduction

Comment 1: Text: CC is the leading cause of cancer-related deaths, with about 5,236 new cases diagnosed annually. 
Comment: suggesting to use number of deaths instead number of new cases since statement pertains to CC as the leading cause of cancer-related mortality

Response 1: Thank you for pointing this out. We agree with this comment. Since there is no latest records on the number of cervical cancer deaths in Kenya, we think added the number of deaths estimated in 2020 and retained the number of approximate number of cacses diagnosed. “In Kenya, CC is the leading cause of cancer-related deaths, with about 5,236 new cases diagnosed annually and over 3,200 deaths in 2002” (page 2, paragraph 1, line 49).

Comments 2: Results: Table 1 - can be placed in supplementary data

Response 2: Thank you for the suggestion. We however decided to retain this table as it provides a quick snapshot of the entire sample. We kept the demographics of the qualitative sub-sample on the supplementary data section.

Reviewer 2 Report

Comments and Suggestions for Authors

The manuscript was well written but some minor improvements are required.

Abstract 

Line5: two Kenyan rural Couties, please confirm if Couties is the intended word or Counties

Introduction

Line 1: Sub-Saharan Africa (SSA) - I detected multiple uses of the abbreviation 'SSA' in the discussion section. You should predefine it at first use.

Line 7: A two-dose HPV vaccination program was .... Please state if Cervarix or Gardasil was used.

Methods

section 2.5.1, Line7: change less or equal to with symbol

Pg6/25: Line 13: or mother])  remove redundant close bracket ']'

Results

Table 1: You should explain the different educational grades in more internationally known terminologies. Eg. Primary, lower secondary, upper secondary, pre-university. Grade 1-3, Grade 4-8 is not readily understood.

Table 3: Column one is too small and obscures the word. Could you consider adjusting the width of Column 1? You should remove the bullets in column 3 and 4.

Author Response

RESPONSES

Summary

Thank you very much for taking the time to review this manuscript. Please find the detailed responses below and the corresponding revisions in the re-submitted files

Point by point responses.

INTRODUCTION

Comments 1: Line5: two Kenyan rural Couties, please confirm if Couties is the intended word or Counties

Response 1: Thank you for pointing this out. This was an error in Spelling of the word “Counties. We have rectified this. “Data were drawn from a mixed-methods study in two Kenyan rural Counties. (page 1, abstract, 3rd sentence)

Comments 2: Line 1: Sub-Saharan Africa (SSA) - I detected multiple uses of the abbreviation 'SSA' in the discussion section. You should predefine it at first use.

Response 2: Thank you for pointing this out. We have, accordingly, predefined the abbreviation in page 2, line 47.

 Comment 3: Line 7: A two-dose HPV vaccination program was .... Please state if Cervarix or Gardasil was used.

Response 3: thank you for the comment. Upon further investigation, we found that the quadrivalent Gardasil-4 was introduced in Kenya in October 2019. We have clarified this on the manuscript : “In 2019, the quadrivalent, two-dose HPV vaccine Gardasil-4, was introduced, targeting adolescent girls and young women in schools and health facilities” (page 2, paragraph 1 lines 53-54)

METHODS

Comment 1: section 2.5.1, Line7: change less or equal to with symbol.

Response 1: “less or equal to” has been changed to (page 5 line 31-32)

Comment 2: Pg6/25: Line 13: or mother]) remove redundant close bracket 'Response 2: close bracket removed (page 6, paragraph 2, line 14)

RESULTS

Comment 1: Table 1: You should explain the different educational grades in more internationally known terminologies. Eg. Primary, lower secondary, upper secondary, pre-university. Grade 1-3, Grade 4-8 is not readily understood.

Response 1: We agree with the comment. We have used both terms to suit international audience. Lower primary (grade 1- 3), and upper primary (grades 4-8); (page 5 line 27; and table 1, page 7 line 27).

Comment 2:  Table 3: Column one is too small and obscures the word. Could you consider adjusting the width of Column 1? You should remove the bullets in column 3 and 4.

Response 2: Thank you for pointing this out. Table 3 columns have been formatted, with column 1 adjusted and bullet points in columns 3 and 4 removed.

Reviewer 3 Report

Comments and Suggestions for Authors

Overall, I think this manuscript has a solid background and purpose of the study. It is evident from the literature where the gaps lie, which relates to the self-sampling willingness to self-collect samples for HPV DNA testing particularly in rural settings. The study has a strong theoretical framework which explains the contextual factors at each levels driving health behaviors for reducing cervical cancer burden among the rural populations in Kenya. 

Although the literature review, and study design has been explained well, highlighting upon the independent and dependent variables, yet it would be useful to state the research questions and hypotheses. 

The authors did an excellent job in explaining the data collection and participant recruitment (with proper use of exclusion and inclusion criteria) by using quantitative, qualitative research methods, integrated with mixed-methods research findings. 

Instruments were clearly mentioned in the study, and the measures were found to be reliable with Cronbach’s alpha. However, I think it needs attention to work on the HELT-LL internal consistency reliability scale 0.57 (which falls below the acceptable range of <0.7)

Findings from this study showed both descriptive and inferential statistical data, provided by the bivariate logistic regression findings explaining the predictor variables explaining the self-sampling willingness as the outcome variable. It is clearly evident from the table how the qualitative and quantitative findings provide recommendations for adoption of HPV-DNA self-sampling, which is a significant finding and strengths of this study.

The discussion section has been detailed with clear explanations interpreting the results, including sharing various study limitations and strengths. I think this study has more limitations over strengths. However, it is not clear about how this practice may work in diverse communities in Kenya, and that might influence adherence on multiple health behaviors. I think the authors could provide a sense of understanding about how participants (say for example low versus higher income in rural communities) may respond to self-sampling willingness for HPV testing, keeping in mind to extend this study into clinical practice and policy standpoint, to support with stronger evidence to intervention planning and implementation.

On a holistic perspective, I would say that this manuscript serves as a foundational study to future studies in clinical settings which is certainly a plus, and a novel contribution to behavioral health among rural populations in Kenya. I think this formative study provided a good foundation on the factors influencing health behaviors at multiple levels, prior to future intervention studies, scaling, and implementation. Just a small note that there are a few errors in spelling/grammar. 

Author Response

Response to Reviewer 4 Comments

1. Summary

Thank you very much for taking the time to review this manuscript. Please find the detailed responses below. Revisions and corrections have been incorporated into the re-submitted files, with corresponding changes indicated by a yellow highlight, page and line numbers.

2. Questions for General Evaluation

Reviewer’s Evaluation

Response and Revisions

Does the introduction provide sufficient background and include all relevant references?

Can be improved

We have added a research question and hypotheses (page. 3 lines 14 to 30)

Are all the cited references relevant to the research?

Is the research design appropriate?

Yes

Not applicable

Are the methods adequately described?

Yes

Not applicable

Are the results clearly presented?

Can be improved

In consideration of the reviewer’s comment about the health literacy variable and the instrument used to assess it having a reliability <0.7, we removed the health literacy variable from our analysis and results. We added the reliability of the cervical cancer stigma scale in this study (0.83); (page 6 lines 21-23).

Are the conclusions supported by the results?

Can be improved

We have eliminated health literacy results from tables 1 (page 7 line 22) and 2 (page 10 line 1). With the addition of the research question and hypotheses (page. 3 lines 14 to 30), our presentation of findings and conclusions are more systematic. We also added results and conclusions based on our hypotheses and findings. Contrary to our hypothesis, healthcare decision-making was not significantly associated with willingness to self-collect samples for HPV-DNA testing, suggesting that other factors—such as access to information and structural barriers—may play a more influential role in shaping screening decisions, regardless of who makes healthcare choices within the household. (page 11 lines 14-16 and page 20; lines 20-24).  

3. Point-by-point response to Comments and Suggestions for Authors

Comment 1: Overall, I think this manuscript has a solid background and purpose of the study. It is evident from the literature where the gaps lie, which relates to the self-sampling willingness to self-collect samples for HPV DNA testing particularly in rural settings. The study has a strong theoretical framework which explains the contextual factors at each levels driving health behaviors for reducing cervical cancer burden among the rural populations in Kenya. 

Response 1: Thank you for the compliment.

Comment 2: Although the literature review, and study design has been explained well, highlighting upon the independent and dependent variables, yet it would be useful to state the research questions and hypotheses. 

Response 2: Research questions and hypotheses have been added (page 3 lines 14-30)

Comment 3: The authors did an excellent job in explaining the data collection and participant recruitment (with proper use of exclusion and inclusion criteria) by using quantitative, qualitative research methods, integrated with mixed-methods research findings. 

Response 3: Thank you for the compliment

Comment 4: Instruments were clearly mentioned in the study, and the measures were found to be reliable with Cronbach’s alpha. However, I think it needs attention to work on the HELT-LL internal consistency reliability scale 0.57 (which falls below the acceptable range of <0.7).

Response 4: Thank you for pointing this out. We found out that the reference cited was outdated and indeed the reliability of the health literacy instrument was lower than the acceptable range. We therefore removed the health literacy variable from our analysis (page5, lines 44-45), and results in tables 1 (page 7 line 22) and 2 (page 10 line 1).

Comment 5: Findings from this study showed both descriptive and inferential statistical data, provided by the bivariate logistic regression findings explaining the predictor variables explaining the self-sampling willingness as the outcome variable. It is clearly evident from the table how the qualitative and quantitative findings provide recommendations for adoption of HPV-DNA self-sampling, which is a significant finding and strengths of this study.

Response 5: Thank you for the compliment

Comment 6: The discussion section has been detailed with clear explanations interpreting the results, including sharing various study limitations and strengths. I think this study has more limitations over strengths. However, it is not clear about how this practice may work in diverse communities in Kenya, and that might influence adherence on multiple health behaviors. I think the authors could provide a sense of understanding about how participants (say for example low versus higher income in rural communities) may respond to self-sampling willingness for HPV testing, keeping in mind to extend this study into clinical practice and policy standpoint, to support with stronger evidence to intervention planning and implementation.

Response 6: Thank you for the compliment

Comment 7: On a holistic perspective, I would say that this manuscript serves as a foundational study to future studies in clinical settings which is certainly a plus, and a novel contribution to behavioral health among rural populations in Kenya. I think this formative study provided a good foundation on the factors influencing health behaviors at multiple levels, prior to future intervention studies, scaling, and implementation.

Response 7: Thank you for the compliment

Comment 8: Just a small note that there are a few errors in spelling/grammar.

Response 8: Thank you for pointing this out. We have reviewed the manuscript in entirety and corrected errors (highlighted in yellow throughout the manuscript).